META-RESEARCH ARTICLE

# Biomedical researchers' perspectives on the reproducibility of research

Kelly D. Cobey[1,2]*, Sanam Ebrahimzadeh[3], Matthew J. Page[4], Robert T. Thibault[5,6], Phi-Yen Nguyen[4], Farah Abu-Dalfa[1,7], David Moher[2,3]

1 University of Ottawa Heart Institute, Ottawa, Canada, 2 School of Epidemiology and Public Health, University of Ottawa, Ottawa, Canada, 3 Centre for Journalology, Ottawa Hospital Research Institute, Ottawa, Canada, 4 Methods in Evidence Synthesis Unit, School of Public Health and Preventive Medicine, Monash University, Melbourne, Australia, 5 Meta-Research Innovation Center at Stanford (METRICS), Stanford University, Stanford, California, United States of America, 6 Coalition for Aligning Science, Chevy Chase, Maryland, United States of America, 7 School of Political Studies, Faculty of Social Sciences, University of Ottawa, Ottawa, Canada

* kcobey@ottawaheart.ca

**Data Availability Statement:** All data and study materials are available on the Open Science Framework (https://osf.io/3ksvz/) or in the supplementary materials.

## Abstract

We conducted an international cross-sectional survey of biomedical researchers' perspectives on the reproducibility of research. This study builds on a widely cited 2016 survey on reproducibility and provides a biomedical-specific and contemporary perspective on reproducibility. To sample the community, we randomly selected 400 journals indexed in MEDLINE, from which we extracted the author names and emails from all articles published between October 1, 2020 and October 1, 2021. We invited participants to complete an anonymous online survey which collected basic demographic information, perceptions about a reproducibility crisis, perceived causes of irreproducibility of research results, experience conducting reproducibility studies, and knowledge of funding and training for research on reproducibility. A total of 1,924 participants accessed our survey, of which 1,630 provided useable responses (response rate 7% of 23,234). Key findings include that 72% of participants agreed there was a reproducibility crisis in biomedicine, with 27% of participants indicating the crisis was "significant." The leading perceived cause of irreproducibility was a "pressure to publish" with 62% of participants indicating it "always" or "very often" contributes. About half of the participants (54%) had run a replication of their own previously published study while slightly more (57%) had run a replication of another researcher's study. Just 16% of participants indicated their institution had established procedures to enhance the reproducibility of biomedical research and 67% felt their institution valued new research over replication studies. Participants also reported few opportunities to obtain funding to attempt to reproduce a study and 83% perceived it would be harder to do so than to get funding to do a novel study. Our results may be used to guide training and interventions to improve research reproducibility and to monitor rates of reproducibility over time. The findings are also relevant to policy makers and academic leadership looking to create incentives and research cultures that support reproducibility and value research quality.

**Funding:** The author(s) received no specific funding for this work.

**Competing interests:** The authors have declared that no competing interests exist.

## Introduction

There is growing interest in both the reproducibility of research and ways to enhance research transparency [1–4]. Terminology around reproducibility varies [5]; here, we define reproducibility as re-doing a study using similar methods and obtaining findings consistent with the original study and as irreproducible when the findings are not consistent with the original study. This definition allows for variation in methods (e.g., conceptual and direct replications) between the original study and the reproducibility study as well as different definitions of how "consistent results" are defined (i.e., using $p$-value, observing results in the same direction, comparing effect sizes). Reproducibility of research is core to maintaining research trustworthiness and fostering translation and progressive discovery. Despite the seemingly critical role of reproducibility of research and growing discussions surrounding reproducibility, the reality is that most studies, including pivotal studies within several disciplines, have never been formally subjected to a reproducibility effort. For example, in education research, an analysis of publications in the field's top 100 journals showed that just 0.13% publications (221 out of 164,589) described reproducibility projects [6]. In psychology, a study examining 250 articles published between 2014 and 2017 found that 5% described a reproducibility effort [7], while a similar study examining reproducibility in social sciences found that just 1% of articles sampled described a focused on reproducing previous results [8]. Across disciplines, our knowledge about the proportion of studies that are reproducible tends to be dominated by a small number of large-scale reproducibility projects. In psychology, for example, a study which estimated the replicability of 100 foundational studies in top journals in the field reported that only 36% had statistically significant results (one measure of reproducibility), compared to 97% of the original studies [9].

In 2016, Nature reported on a survey of more than 1,500 researchers about their perceptions of reproducibility. They found that 83% agreed there was a reproducibility crisis in science, with 52% indicating that they felt the crisis was "significant" [10]. Survey studies like this play a powerful role in elucidating the determinants of reproducibility. Such information is essential to identify gaps in factors including training, research support, and incentives to ensure reproducible research. Given the global nature of research, capturing global perspectives as was done in the Nature survey is crucial to obtaining broad understanding of issues across the research ecosystem.

In this study, we aim to build on the 2016 Nature survey on reproducibility by surveying researchers in the biomedical community specifically. There is immediate importance to ensuring biomedical research is reproducible: here, studies that were subsequently not reproducible have led to patient harms [11,12]. By capturing a diverse and global group of biomedical researchers' perceptions of reproducibility within the field we hope to better understand how to ensure reproducibility in biomedicine. While our work is inspired by the 2016 Nature survey, this is not a direct effort to reproduce that study: we sample a different community and use a sampling approach that differs from the original study which limits the ability for direct comparison. Our specific objectives were to: (1) explore biomedical researchers' perceptions of reproducibility and their perceptions of causes of irreproducibility; and (2) describe biomedical researchers' experiences conducting and publishing reproducibility projects. The study is descriptive, so we had no formal hypotheses. Understanding researcher perceptions is a key starting point to drive culture change and to create training interventions to drive improvement.

## Methods

### Open science statement

This study received ethics approval from the Ottawa Health Sciences Research Ethics Board (20210856-01H). This study protocol was posted a priori, and data and materials have been made available: https://osf.io/3ksvz/ [13].

### Study design

We conducted an online cross-sectional closed survey of researchers who published a paper in a journal indexed in MEDLINE (RRID:SCR_002185). The survey was anonymous.

### Sampling framework

We downloaded the MEDLINE database of journals. From this list of approximately 30,000 journals, we selected a random sample of 400 journals using the RAND() function in Excel (RRID:SCR_016137). We then extracted author names and emails from all articles published in those journals between October 1, 2020 and October 1, 2021. We included all authors whose names and emails were available and all article types/study designs. For full details on our semi-automated approach to extracting author emails, please see our search strategy in S1 File.

### Participant recruitment

The survey was sent only to those researchers identified via our sampling procedure (i.e., a closed survey). Potential participants received an email containing a recruitment script which detailed the purpose of the study and invited them to review our informed consent form and complete our anonymous online survey. Participation in the survey served as implied consent; we did not require signed consent to maintain anonymity. To send emails to the sample of authors, we used mail merge feature in Microsoft 365. This tool allows for the personalization of emails without having to individually customize and send each out. In the case of non-response, we sent 3 reminder emails to potential participants at weekly intervals after the initial invitation. We closed the survey 4 weeks after the initial invitation. We did not provide any specific incentive to complete the survey.

### Survey

The full survey is available in S2 File. The survey was administered using SurveyMonkey and could be completed in about 10 min. The survey contained a total of 19 questions starting with 4 demographic questions about the participants, including their gender, research role, research area, and country of residence. Participants were then asked to complete questions about their perceptions of reproducibility in biomedicine, questions about their experience with reproducibility, and questions about perceptions of barriers and facilitators to conducting reproducibility projects. The survey questions were presented sequentially and using adaptive formatting to present only certain items based on the participant's response to previous questions. Most questions were multiple choice, with 2 questions asking participants to expand on their responses using a free-text box. The survey was purpose-built for the study by the research team, building directly off the previously published Nature reproducibility survey [10]. We included several of the previous study's questions directly in this study, modified some slightly, and made some more specific to the biomedical research setting. We also introduced some novel questions on reproducibility. The survey was pilot tested by 3 researchers (not on the team) to ensure clarity and acceptability of format and we edited the survey to

address their feedback regarding the clarity of survey questions and issues with illogical survey flow. Participants were able to skip any question.

### Data management and analysis

Data were exported from SurveyMonkey and analyzed using SPSS 28 (RRID:SCR_002865). We report descriptive statistics including count and percentages for all quantitative items. For the qualitative items, we conducted a thematic content analysis. To do so, 2 researchers individually read all text-based responses and assigned a code to summarize the content of the text. This inductive coding approach involves creating codes based on the data itself, rather than from a pre-established coding framework. Codes were refined iteratively upon exposure to each text-based response read. After discussion to reach consensus on the codes used, we then grouped the agreed codes into themes for reporting in tables. Coders were members of the project team who were not blinded to the study aims.

## Results

### Protocol amendments

In our original protocol, we said we would take a random sample of 1,000 journals from MEDLINE and extract information from the first 20 authors. This approach required extensive manual extraction, so we opted to restrict our random sample to 400 journals and semi-automate extraction of author information for an entire year's worth of publications. Our revised method meant that we obtained and used listed emails from all authors on an identified paper (i.e., we were not restricted to corresponding authors).

### Demographics

A total of 24,614 emails were sent, but bounce backs were received from 1380, meaning 23,234 emails were sent successfully to potential participants. A total of 1,924 participants accessed our survey, of whom 1,630 participants provided completed responses (response rate 7%; this frequency is slightly lower than the estimated 1,800 responses reported in our protocol). Most participants were Faculty Members/Primary Investigators ($N = 1,151$, 72%) and more than half of participants were male ($N = 943$, 59%). Respondents were from more than 80 countries, with the USA ($N = 450$, 28%) having the highest representation. About half of participants reported working in clinical research ($N = 819$, 50%). Further demographic details by role, gender, country, and research area are provided in Table 1.

### Perceptions of reproducibility

When asked whether there was a reproducibility crisis in biomedicine most researchers agreed ($N = 1,168$, 72%), with 27% ($N = 438$) indicating the crisis was significant and 45% ($N = 703$) indicating a slight crisis (we note that a "slight crisis" is a bit of an oxymoron, but retained the wording from the original Nature survey for comparison purposes); see Fig 1 and S3 File for breakdown by discipline. Compared to the previously published Nature study ($N = 819$, 52%), fewer participants in our study felt there was a "significant reproducibility crisis" ($N = 438$, 27%). This difference was even larger when we restricted to the Nature study participants who indicated they worked in medicine ($N = 203$, 60%).

Participants were then asked what percentage of papers in each of biomedical research overall, clinical biomedical research, in vivo biomedical research, and in vitro biomedical research they thought were reproducible, see Fig 2. Only 5% ($N = 77$) thought more than 80% of biomedical research was reproducible. See S3 File for complete results. We provide a

**Table 1. Participant demographics.**

| Item | Response options | N | % |
|---|---|---|---|
| Researcher role | Graduate student | 88 | 6 |
| | Postdoctoral fellow | 129 | 8 |
| | Faculty member/PI | 1,151 | 72 |
| | Research support staff | 54 | 3 |
| | Scientist in industry | 28 | 2 |
| | Scientist in third sector | 27 | 2 |
| | Government scientist | 54 | 3 |
| | Other | 73 | 5 |
| | *Missing data* | 26 | - |
| Gender | Female | 643 | 40 |
| | Male | 943 | 59 |
| | Non-binary | 3 | 0.2 |
| | Prefer to self-describe | 1 | 0.1 |
| | Prefer not to say | 13 | 1 |
| | *Missing data* | 27 | - |
| Country of Employment (Top 3) | USA | 450 | 28 |
| | Canada | 128 | 8 |
| | UK | 105 | 7 |
| | *Missing data* | 32 | - |
| Research Area | Clinical research | 819 | 50 |
| | Preclinical research–in vivo | 191 | 12 |
| | Preclinical research–in vitro | 163 | 10 |
| | Health systems research | 147 | 9 |
| | Methods research | 81 | 5 |
| | Other, please specify | 227 | 14 |
| | *Missing data* | 2 | - |

breakdown of responses between genders, between researchers in different biomedical research areas, and by career rank in S4 File.

## Determinants of irreproducibility

When presented with various potential causes of irreproducibility, more than half of participants responded that each presented factor contributes to irreproducibility. The top characteristic participants noted as "always contributing" to irreproducibility was pressure to publish ($N = 300$, 19%). Factors deemed least likely to contribute to irreproducibility were fraud ($N = 320$, 20%) and bad luck ($N = 568$, 36%). See Table 2 for complete results.

A total of 97 (6%) participants provided a written response to elaborate on what they perceived were causes of irreproducibility. Responses were coded into 16 unique codes and then thematically grouped into 7 categories: ethics, research methods, statistical issues, incentives, issues with journal and peer review, lack of resources, and other. For definitions and illustrative examples of the codes, see Table 3.

## Experiences with reproducibility

Participants were asked about their experience conducting reproducibility projects. Nearly a quarter of participants indicated that they had previously tried to replicate one of their own published studies and failed to do so ($N = 373$, 23%), whereas 31% ($N = 501$) indicated all such

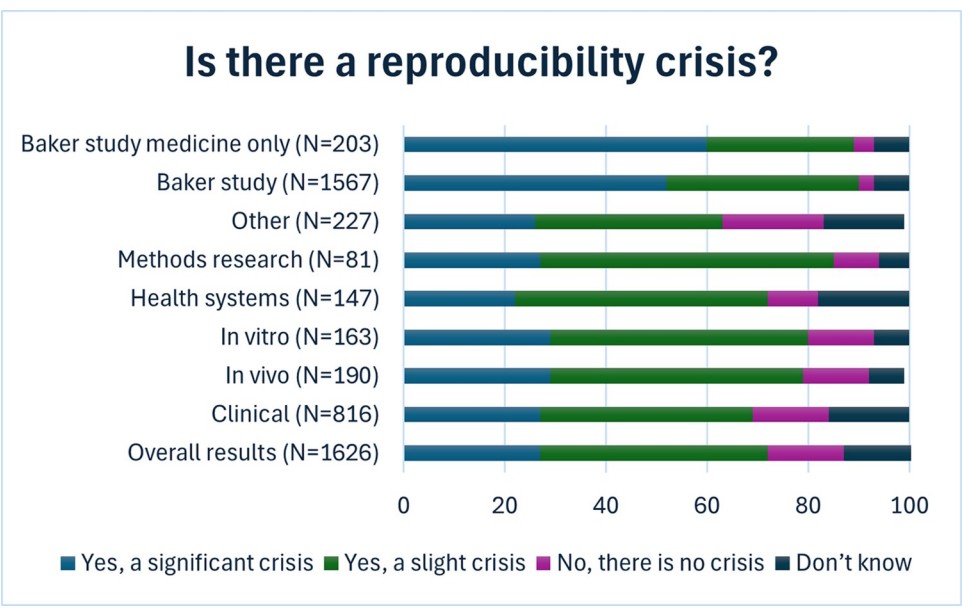

**Fig 1. Participant perceptions of a reproducibility crisis.** Data is presented overall for all participants in the current study and is broken down by research focus area in medicine. Results are presented in context to the overall Nature study findings and specifically to participants from this study indicating they worked in medicine. The underlying data for this figure can be found at https://osf.io/dbh2a.

replications they had conducted had yielded the same result as the original, and slightly less than half of participants indicated that they had never tried to replicate any of their own published work (N = 734, 46%). Among the 874 participants who indicated they had tried to replicate one of their own studies, when asked to consider their most recent replication effort, 313 (36%) indicated they had published the results.

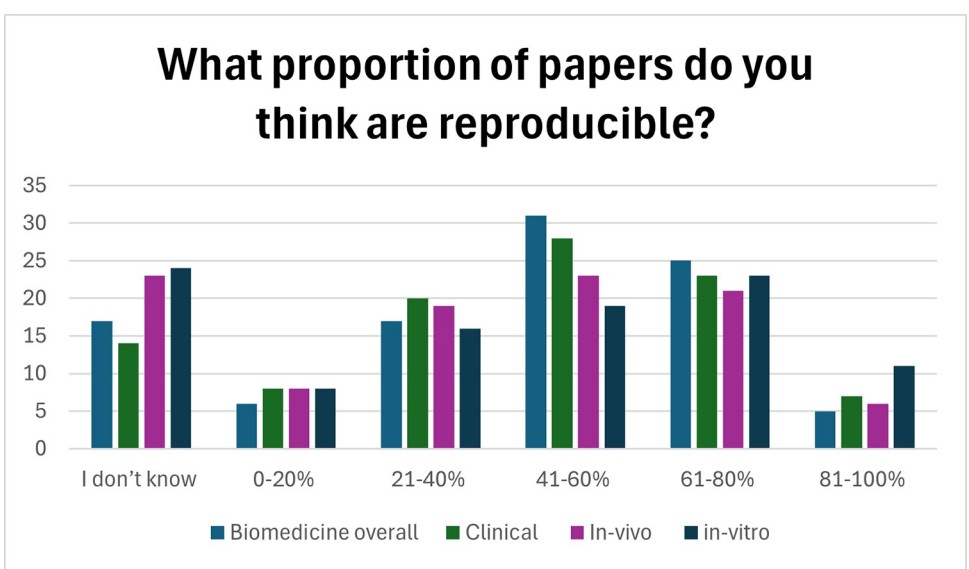

**Fig 2. Participants perceptions of the proportion of papers they think are reproducible in biomedicine overall and by biomedical research area.** The underlying data for this figure can be found at https://osf.io/dbh2a.

**Table 2. Participant perceptions of the causes of irreproducibility.**

| | N(%) | | | | | |
| --- | --- | --- | --- | --- | --- | --- |
| | Always contributes | Very often Contributes | Sometimes Contributes | Does not Contribute | Unsure | Missing data |
| Selective reporting of the published literature | 131 (8) | 638 (40) | 714 (45) | 43 (3) | 73 (5) | 31 |
| Selective publication of entire studies | 182 (11) | 698 (44) | 577 (36) | 71 (4) | 71 (4) | 31 |
| Pressure to publish | 300 (19) | 693 (43) | 473 (30) | 75 (5) | 57 (4) | 32 |
| Low statistical power | 185 (12) | 706 (44) | 579 (36) | 76 (5) | 48 (3) | 36 |
| Poor statistical analysis | 197 (12) | 615 (38) | 649 (41) | 99 (6) | 44 (3) | 26 |
| Not enough internal replication (E.g., by the original lab/authors) | 132 (8) | 539 (34) | 697 (44) | 93 (6) | 142 (9) | 27 |
| Insufficient study oversight | 86 (5) | 376 (24) | 799 (50) | 194 (12) | 143 (9) | 32 |
| Lack of training in reproducibility | 153 (10) | 522 (33) | 622 (39) | 168 (11) | 135 (8) | 30 |
| Failure to make materials openly available | 141 (9) | 449 (28) | 722 (45) | 191 (12) | 99 (6) | 28 |
| Failure to make original study data openly available | 137 (9) | 476 (30) | 685 (43) | 205 (13) | 94 (6) | 33 |
| Poor study design | 208 (13) | 584 (36) | 678 (42) | 96 (6) | 38 (2) | 26 |
| Fraud | 185 (12) | 120 (8) | 624 (40) | 320 (20) | 330 (21) | 51 |
| Poor quality peer review | 140 (9) | 437 (27) | 755 (47) | 192 (13) | 72 (5) | 34 |
| Problems in the design of replication studies | 103 (6) | 406 (25) | 809 (51) | 162 (10) | 123 (8) | 27 |
| Technical expertise required for replication | 96 (6) | 429 (27) | 743 (46) | 190 (12) | 144 (9) | 28 |
| Variability of standard reagents | 82 (5) | 288 (18) | 617 (39) | 229 (14) | 380 (24) | 34 |
| Bad luck | 23 (1) | 70 (4) | 461 (29) | 568 (36) | 466 (29) | 42 |

Almost half of the participants indicated that they had previously tried to replicate a published study conducted by another team and failed to do so ($N = 724$, 47%), whereas 10% ($N = 156$) indicated all replications that they had attempted were successful, while 43% ($N = 666$) indicated they had never tried to replicate someone else's published research. Among those who had published their replication study, when asked to consider their most recent replication effort, 29% ($N = 224$) indicated it took about the same amount of time to publish as a typical non-replication paper. A quarter ($N = 189$, 25%) of participants who had attempted to replicate others' research indicated they had no plans to publish their replication study. Eighty-five percent of participants ($N = 1,316$) indicated they had never been contacted by another researcher who was unable to reproduce a finding they previously published. See Table 4 for complete results.

A total of 724 participants responded to the item about why they replicated their own research, of which 675 (93%) provided relevant text-based responses. Responses were coded into 17 unique codes and then grouped into 7 themes: training, ensuring reproducibility, additional research, addressing concerns, joining/setting up a new lab, for publication or due to peer review, and other. For illustrative examples of the codes, see Table 5.

A total of 748 participants responded to the item about why they replicated someone else's research, of whom 700 (94%) provided relevant text-based response data. Responses were coded into 19 unique codes and then grouped into 7 themes: trustworthiness, extending and improving research, application to new setting, new research, interest, training, and other. For illustrative examples of the codes, see Table 6.

## Support for initiatives to enhance reproducibility

Few participants reported that their research institution has established procedures to enhance reproducibility of biomedical research ($N = 254$, 16%), and almost half reported that their

**Table 3. Thematic analysis of perceived causes of irreproducibility.**

| Themes | Codes | N (97) | % | Example |
|---|---|---|---|---|
| Ethics | Conflicts of interest | 3 | 3 | "Conflicts of interest, commercial interests, corporate interests" |
| | Fraud | 2 | 2 | "Sure sometimes there is fraud or poor study design or execution. . ." |
| Research methods | Complex research design or methods | 2 | 2 | "specialized or cutting edge techniques not adopted or fully appreciated by enough other labs" |
| | Heterogeneity in biology/ environment | 25 | 26 | "heterogeneity of included subjects" |
| | Lack of standard methods | 6 | 6 | "lack of precise outcome measures for clinical studies" |
| | Poor study design or planning | 8 | 8 | "Poor design of original studies with increased Type 1 error due to multiple comparisons/endpoints" |
| Statistical issues | Discretion in statistical analysis | 5 | 5 | "Investigators conducting their own analyses" |
| | Overreliance on statistics | 3 | 3 | "over interpretation of statistics—0.05 p value without thinking enough about methods behind it and meaning" |
| | sample size/power issues | 4 | 4 | "Small effects of biomedical phenomena" |
| Incentives | Lack of value for reproducibility studies | 4 | 4 | "no incentive to reproduce studies" |
| | Preference for novelty | 10 | 10 | ". . .They are so fixated on novelty they absolutely discourage replication and/or the publishing alternative findings." |
| | Pressure to publish | 2 | 2 | "I think it is the pressure to publish.." |
| | Researcher attitudes | 2 | 2 | "Preconceptions of investigators and reviewers.." |
| Issues with journals and peer review | Issues with journals and peer review | 22 | 23 | "Journals not accepting replication studies." |
| Lack of resources | Lack of resources | 14 | 14 | "Almost no funding for replication studies" |
| Other | Other | 6 | 6 | "Desire to have a convincing story, results" |

institutions did not provide training on how to enhance the reproducibility of research ($N = 731$, 48%) with an additional 503 (33%) reporting they were unsure of whether such training existed. We asked participants to provide information and links to relevant reproducibility training at their institution, which resulted in information or (functioning) links to 24 unique training resources (see 5). Among these 24 sources, just 9 (38%) clearly described specific openly available (e.g., no paywall) training related to reproducibility. Most researchers responded that they perceived their institution would value them doing new biomedical research studies more than replication studies ($N = 1031$, 67%). The majority also indicated that it would be harder to find funding to conduct a replication study than a new study ($N = 1,258$, 83%), with just 7% ($N = 112$) indicating they were aware of funders providing specific calls for conducting reproducibility related research. For full results, see Table 7.

We asked participants to indicate how much they agreed with the statement "I feel I am more concerned about reproducibility than the average researcher in my field and at my career stage" as a way to indirectly address potential bias in self-selection to complete the survey. Participants responded on a 5-point scale with endpoints, strongly disagree (1) and strongly agree (5). Participants reported a mean response of 3.2 ($N = 1,402$, SD = 0.89) that corresponds to the mid-point of the scale "neither agree nor disagree."

## Discussion

We report the results of an international survey examining perceptions of reproducibility. Almost 3 quarters of participants reported that they felt there was a reproducibility crisis in biomedicine. The concern appears to apply to biomedicine overall, but also specifically to clinical research, in vivo research, and in vitro research (11% or fewer participants indicated that

**Table 4. Participant experiences with reproducibility.**

| Item | Response options | N | % |
|---|---|---|---|
| Have you ever tried to replicate a published study <u>you</u> previously conducted and failed? | Yes | 373 | 23 |
| | No- all replications I have completed of my own research have been successful | 501 | 31 |
| | No – I have never tried to replicate my own research | 734 | 46 |
| | Missing data | 22 | - |
| Did you publish your replication study results of your study? | Yes – but it took longer to publish than other papers you've published that were not replications | 139 | 17 |
| | Yes – and it took about the same amount of time to publish as other papers you've published that were not replications | 152 | 19 |
| | Yes – but it was quicker to publish than other papers you've published that were not replications | 22 | 3 |
| | No – I have submitted but not yet had the work accepted | 29 | 4 |
| | No – I have not yet submitted, but intend to do so | 84 | 10 |
| | No – I don't intend to attempt to publish this study No – Journals don't appear interested in publishing replications | 205117 | 2514 |
| | Other | 75 | 91 |
| | Missing data | 51 | - |
| Have you ever tried to replicate a published study conducted by <u>another team</u> of authors and failed? | Yes | 724 | 47 |
| | No- all replications I have completed have been successful | 156 | 10 |
| | No – I have never tried to replicate someone else's published researchMissing | 66684 | 43- |
| Did you publish your replication study results of the other researchers' study? | Yes – but it took longer to publish than other papers you've published that were not replications | 139 | 18 |
| | Yes – and it took about the same amount of time to publish as other papers you've published that were not replications | 224 | 29 |
| | Yes – but it was quicker to publish than other papers you've published that were not replications | 14 | 2 |
| | No – I have submitted but not yet had the work accepted | 21 | 3 |
| | No – I have not yet submitted, but intend to do so | 79 | 10 |
| | No – I don't intend to attempt to publish this study | 189 | 25 |
| | Don't appear interested in publishing replications | 56 | 7 |
| | Other | 43 | 6 |
| | Missing data | 12 | - |
| Have you ever been contacted by another researcher who was unable to reproduce a finding you published? | Yes | 164 | 11 |
| | No I can't remember | 131653 | 853 |
| | Unsure | 16 | 1 |
| | Missing data | 81 | - |

they think more than 80% of papers in each category were reproducible). Researchers agreed that a variety of factors contribute to irreproducibility; however, the chief characteristic that most participants indicated "always contributes" to irreproducible research was a pressure to publish. Concerns about how the current system of academic rewards stresses quantity over quality have been expressed for decades [14–16]—a sentiment supported by this study's data,

**Table 5. Thematic analysis of reasons why participants replicated their own study.**

| Theme | Code | N | % | Example |
|---|---|---|---|---|
| Training | Education purposes | 15 | 2 | "Used it for teaching purposes" |
| | New students/staff replicate former results | 27 | 4 | "To teach new batch of PhD students and ask them replicate senior students experiments" |
| Ensuring reproducibility | Could not reproduce a finding so studied why | 15 | 2 | "Students reported difficulty replicating original methods" |
| | Findings were challenged | 9 | 1 | "Others have reported opposite findings so we wanted to verify our results." |
| | Interesting | 8 | 1 | "I was curious." |
| | Replication is part of research approach norms | 22 | 3 | "As a clinical researcher, we repeat some evaluations over time to detect modified trends." |
| | To confirm/validate the finding | 309 | 46 | "To check if finding were comparable over time" |
| | Public/community value | 6 | 1 | "This is very important for public acceptability." |
| Additional research | To use as controls | 33 | 5 | "As controls in a follow-up study." |
| | Novel research | 17 | 3 | "Test new ideas" |
| | Extension of study | 232 | 34 | "Replication and extension." |
| Address potential concerns with original study | To address limitations of the original study | 30 | 4 | "Confirmation with larger sample size after pilot proof of concept study" |
| | To address heterogeneity of biology/environment | 35 | 5 | "Search for minor variations in population and results" |
| | Improving quality | 11 | 2 | "Improve the research" |
| Joining/setting up a new lab | Joining/setting up a new lab | 8 | 1 | "Started independent lab" |
| For publication or due to peer review | For publication or due to peer review | 11 | 2 | "Journals request it" |
| Other | Other | 30 | 4 | "Scientific ethics" |

**Table 6. Thematic analysis of reasons why participants replicated someone else's study.**

| Theme | Definition | Codes | N (874) | % |
|---|---|---|---|---|
| Trustworthiness | Expressing doubt and seeking to verify the trustworthiness of the original results | To verify the results | 158 | 18 |
| | | Wary of the original result | 59 | 7 |
| | | Obtained conflicting results | 52 | 6 |
| | | To follow-up/challenge findings | 12 | 1 |
| Extending and improving research | Intending to extend and improve the original study | To extend current research | 129 | 15 |
| | | Ability to extend study with improved methods | 48 | 6 |
| | | Interested in the same question | 26 | 3 |
| | | To provide a more current replication | 16 | 2 |
| | | Already running the same study | 11 | 1 |
| | | Collaborating with original team | 10 | 1 |
| Application to new setting | Intending to apply the original study in a broader or new setting | Replication in a new setting/population | 116 | 13 |
| | | To determine generalizability | 8 | 1 |
| New research | Utilizing the original study and/or studies method in new projects | Wanted to use the new studies method | 73 | 8 |
| | | Reproduced research for new projects | 30 | 3 |
| | | To use as baseline or control data | 23 | 3 |
| Interest | Replicating the original study out of personal interest and/or curiosity | Out of interest/curiosity | 34 | 4 |
| Training | Replicating the original study for the purpose of understanding and gaining new knowledge | To understand the methods better | 17 | 2 |
| | | For educational purposes/to gain new knowledge | 16 | 2 |
| Other | Other | Other | 36 | 4 |

**Table 7. Participants perceived support for reproducibility.**

| Item | Response options | N | % |
|---|---|---|---|
| Does your research institution have established procedures to enhance reproducibility of biomedical research? | Yes | 254 | 16 |
| | No | 655 | 42 |
| | I do not know | 637 | 41 |
| | Missing data | 84 | - |
| My institution would value me doing new biomedical studies more than me doing replication studies. | True | 1,031 | 67 |
| | False | 147 | 10 |
| | I do not know | 351 | 23 |
| | Missing data | 101 | - |
| In my biomedical research setting, it would be harder to find funding to conduct a replication study than it would be to find funding for a new study. | True | 1,258 | 83 |
| | False | 72 | 5 |
| | Unsure | 194 | 13 |
| | Missing data | 106 | - |
| Are you aware of funders providing specific calls for conducting reproducibility related research? | Yes | 112 | 7 |
| | No | 1,417 | 93 |
| | Missing data | 101 | - |
| Does your research institution provide training on how to enhance the reproducibility of research? | Yes, and I have taken it | 184 | 12 |
| | Yes, but I have not taken it | 102 | 7 |
| | No | 731 | 48 |
| | Unsure | 503 | 33 |
| | Missing data | 110 | - |

which suggests that researchers' performance is negatively impacted, in terms of producing reproducible research, by what the academic system incentivizes.

More than half of participants reported having tried to replicate their own work previously, with almost a quarter indicating that when they did so they failed, and many indicating that they do not intend to publish their findings. Similar findings were reported when asked about whether participants had tried to replicate another researcher's study, with 57% indicating they had done so, and 47% indicating the replication failed. The majority of participants had not been contacted by another researcher who was unable to reproduce their findings, which suggests that teams of researchers attempting to reproduce studies do not typically communicate despite the potential value for this contact to enhance reproducibility [9,17].

We asked several items about researchers' perceptions of their institution's support for reproducibility (Table 7) and the findings collectively suggest gaps in incentives and support to pursue reproducibility projects. For example, with just 16% of respondents reporting awareness that their institution has established procedures to enhance reproducibility, and 67% of researchers perceiving their institution values novelty over replication, our results suggest that overall, researchers perceive that institutions are not doing their part to effectively support and reward research reproducibility [18]. The growth of "Reproducibility Networks," national peer-led consortiums aiming to promote reproducibility and understand factors related to irreproducibility, are a promising opportunity to rectify this situation [19]. For example, the UK Reproducibility Network (UKRN) provides an opportunity for institutions to formally commit to reproducibility by joining the network and requires a formal role within the senior management team of each member institution to support the delivery of network activities with the institution. The UKRN boasts a range of events and shared resources to support reproducibility [20]. The structure of reproducibility networks allows for harmonization but

also flexibility in approach. Obtaining a sustained funding mechanism will be critical to their growth and ongoing success in terms of their long-term impact.

This study built off of an earlier study of more than 1,500 researchers surveyed by Nature about reproducibility [10]. The current study differs in several important ways. Firstly, the focus is exclusively on biomedicine, since to our knowledge no large scale and representative survey of biomedical researchers has been conducted to date. Indeed, just 203 (13%) of the 1,576 researchers who completed the original study indicated "medicine" as their main area of interest. Secondly, we randomly sampled researchers from publication lists, meaning we are able to report a response rate. This was not possible in the Nature survey, which was emailed to Nature readers and advertised on affiliated websites and social media outlets, meaning that the number of individuals encountering the survey is unknown. While it is possible that among those invited to take part there is bias among participants who choose to complete the survey, our approach has been chosen to help minimize surveying those explicitly active in reproducibility projects or related initiatives. Indeed, the finding that overall participants report not to differ from their belief of their peers regarding their level of concern about reproducibility provides some assurance that our sampling strategy was effective. We also find there is not much difference between different groups' responses (S4 File).

We acknowledge several limitations in our approach. Firstly, our study survey was purpose-built, and while drawing on the Nature study, was not designed according to a particular theoretical framework. This type of approach lends itself well to the exploratory and descriptive goals of the current study; however, it makes it harder to test hypotheses or measure changes over time. A validated scale to assess perceptions of reproducibility and its cause would be of value to the community for future research. We did not look at correlations between responses to items on our survey, as doing so would be a post hoc change from our protocol. Future research using a validated scale may be better positioned to consider potential relationships between responses a priori (i.e., how one's perception of a reproducibility crisis relates to the perceived proportion of research that is reproducible). Future surveys would also benefit from a more extensive approach to piloting for clarity and readability. Another potential limitation is that the coders conducting the thematic analysis were not blind to the study's objective (i.e., were members of the research team). While the study is descriptive it remains possible that knowledge of the study aim and expertise in the reproducibility literature biased the coding approach. Moreover, only a small number of participants (6%) provided text-based descriptions to nuance their survey responses. It is perhaps not surprising given that the survey offered no incentive to take part, but it raises the possibility that those that took the time to respond to text-based items may differ from those who did not. Indeed, the top 3 most represented countries in our sample overall were the US, Canada, and the UK. While we had over 80 countries represented, our responses tended to be Western-centric which may present another bias, even in these countries tend to be among the most productive producers of research. Future research delving more specifically into recruiting from a specific country, or set of countries, could help improve our understanding of how national contexts may lead to differences in perceptions and practices related to reproducibility. Finally, our sampling approach allowed us to obtain any listed emails on a given study article. This means that more than one of the authors of a given article in our random sample may have responded to the survey. While our random sampling approach provided a strong basis for sampling, this nuance presents the possibility for some bias as authors of different studies may vary more considerably and we have no way of knowing how survey participants were represented within or between given papers. While it may be tempting to conduct comparison testing of our results compared to the earlier Nature replication survey, we are not positioned to do so. Our study did not aim to replicate the original Nature paper and was registered as a descriptive

study. Given that we sampled exclusively biomedical researchers, and that our recruitment approach for sampling differed significantly, it would be very difficult to disentangle whether differences in results reflect sampling bias in the original study, temporal changes in the research ecosystem over time, different perceptions in biomedicine compared to research more broadly, or a combination of these factors. We know of no relevant framework that would have been appropriate to implement in order to test specific hypotheses within the biomedical community. Further, null hypothesis testing is not appropriate for determining beliefs and when hypothesis testing is used you need a formal sample size calculation [21,22].

Concerns about reproducibility are being widely recognized within research but also more broadly in the media [23]. The COVID-19 pandemic has shifted our thinking about research transparency [24] and highlighted issues with fraud in research, poor-quality reporting, and poor-quality study design [25,26], all factors that can contribute to irreproducible research. As stakeholders work to introduce training and interventions to enhance reproducibility, it will be critical to monitor the effectiveness of these interventions. This international survey provides a contemporary cross-section of the biomedical community. While our survey approach and direction of findings are consistent with the previous Nature study, ongoing monitoring of perceptions of reproducibility in the community is critical to gauge shifts over time. Indeed, conducting this same survey again in the future would allow for a temporal comparison on how perceptions and behaviors shift over time. This suggestion aligns with the approach of the EU Commission who in their report "Assessing the reproducibility of research results in EU Framework Programmes for Research" [27], describe an approach to understand and monitor the progress of reproducibility over time. In the work, researchers were also asked about their perceptions of reproducibility and 60% agreed that there is a reproducibility crisis, but also overwhelmingly agreed (82%) that reproducibility is "important" or "very important" in their discipline. Like our findings, participants indicated several key challenges to achieve reproducibility, with issues around cultural factors being paramount. Our finding that a pressure to publish was most likely to be rated as "always contributing" to irreproducibility is consistent with the conclusions of this report. Collectively, the outcomes of this body of evidence highlight perceived causes and constraints to producing reproducible research which should be prioritized within the biomedical community and targeted with interventions and supports to create improvements over time. An important consideration of future survey or reproducibility monitoring will be to be explicit about the language and definitions of reproducibility used —terminology in this space is complex and the same words can reflect different concepts between researchers within and between research areas. Clear definitions for reproducibility will help to ensure that differences observed in survey responses over time, or between groups, reflect more than simply differences in how surveys are interpreted.

## Supporting information

**S1 File. Search strategy used to obtain a random sample of journals to extract author contact information from to identify possible participants.**
(DOCX)

**S2 File. Survey administered to participants.** The survey was administered online and used survey logic to present relevant items.
(DOCX)

**S3 File. Supplementary tables presenting a comparison of perceptions of reproducibility between our findings and the original Nature paper by Baker, presented overall and by discipline.** In addition, we present participant perceptions of reproducibility in different research

areas.
(DOCX)

**S4 File. Additional analyses of survey responses by gender and academic role.**
(DOCX)

**S5 File. List of training available on reproducibility mentioned by survey participants.**
(DOCX)

## Author Contributions

**Conceptualization:** Sanam Ebrahimzadeh, Matthew J. Page, Phi-Yen Nguyen, David Moher.

**Data curation:** Kelly D. Cobey, Sanam Ebrahimzadeh, Phi-Yen Nguyen.

**Formal analysis:** Kelly D. Cobey, Phi-Yen Nguyen, Farah Abu-Dalfa.

**Investigation:** Kelly D. Cobey, Sanam Ebrahimzadeh, Matthew J. Page, Robert T. Thibault, David Moher.

**Methodology:** Kelly D. Cobey, Sanam Ebrahimzadeh, Matthew J. Page, Robert T. Thibault, Phi-Yen Nguyen, David Moher.

**Project administration:** Kelly D. Cobey, Phi-Yen Nguyen.

**Supervision:** Kelly D. Cobey.

**Validation:** Kelly D. Cobey, Sanam Ebrahimzadeh.

**Writing – original draft:** Kelly D. Cobey.

**Writing – review & editing:** Kelly D. Cobey, Sanam Ebrahimzadeh, Matthew J. Page, Robert T. Thibault, Phi-Yen Nguyen, Farah Abu-Dalfa, David Moher.

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
