## [Editor Report · Decision Letter 0]

19 Dec 2023

Dear Kelly, 

Thank you for submitting your manuscript entitled "Biomedical researchers’ perspectives on the reproducibility of research: a cross-sectional international survey" for consideration as a Meta-Research Article by PLOS Biology.

This revised version of your manuscript has now been evaluated by the PLOS Biology editorial staff and I'm writing to let you know that we would like to send your submission out for external peer review. Note that, given the upcoming holidays, it is likely that we will not invite reviewers until January.

Once your full submission is complete, your paper will undergo a series of checks in preparation for peer review. After your manuscript has passed the checks it will be sent out for review. To provide the metadata for your submission, please Login to Editorial Manager (https://www.editorialmanager.com/pbiology) within two working days, i.e. by Dec 21 2023 11:59PM.

Kind regards,

Roli

Roland Roberts, PhD

Senior Editor

PLOS Biology

rroberts@plos.org

---

## [Decision Letter · Decision Letter 1]

21 Mar 2024

Dear Kelly,

Thank you for your patience while your manuscript "Biomedical researchers’ perspectives on the reproducibility of research: a cross-sectional international survey" was peer-reviewed at PLOS Biology. It has now been evaluated by the PLOS Biology editors, an Academic Editor with relevant expertise (this is a different Academic Editor from the one who saw the initial submission), and by three independent reviewers. 

You'll see that reviewer #1 says that this is well-written and highly relevant, but is not fully convinced by the novelty with respect to the Baker study, or the design of the work. S/he finds it confirmatory and descriptive (lacking in hypotheses), wonders whether the design reduces the risk of bias, and thinks that the Discussion fails to move us forward. Although in balance negative, their review seems helpful and constructive. Reviewer #2 says that he could not find the supplementary materials (he’s right; these are missing), suggests some presentational improvements (better Table structure, heatmaps), asks for some additional analysis (correlation analyses), wants you to declare some limitations, asks about the distinction between replicability and reproducibility, and asks you to provide code. Reviewer #3 is the most explicitly positive, but wants much better presentation (“graphs, figures, colour” – there are currently no Figures, just Tables), and wants breakdown by country and career stage (she thinks it’s rather “western-centric”). Like rev #2, she thinks the piloting was rather limited and wants you to supply the code.

IMPORTANT: We discussed these comments with the Academic Editor, who agreed that we should invite a revision, saying "Although the study was built on Baker (2016), data are original because time went by and many new discussions and actions happened in the meanwhile. I found no problem with the qualitative approach when it is suitable to answer the research question, as is the case in this paper. However, I agree with reviewers that limitations should be discussed. Sure, data visualization needs improvement, maybe, combining table with plots."

IMPORTANT: I should add, from the point of view of the journal, that for our broad readership, to enhance the appeal and traction of the paper, improved dataviz is ESSENTIAL, and we strongly support reviewer #3's helpful advice on this point. Also all supplementary files must be supplied, and underlying data and code must be made available, in line with our data and code availability policies.

In light of the reviews, which you will find at the end of this email, we would like to invite you to revise the work to thoroughly address the reviewers' reports.

Given the extent of revision needed, we cannot make a decision about publication until we have seen the revised manuscript and your response to the reviewers' comments. Your revised manuscript is likely to be sent for further evaluation by all or a subset of the reviewers.

**IMPORTANT - SUBMITTING YOUR REVISION**

*Re-submission Checklist*

*Published Peer Review*

*PLOS Data Policy*

Sincerely,

Roli

Roland Roberts, PhD

Senior Editor

PLOS Biology

rroberts@plos.org

REVIEWERS' COMMENTS:

Reviewer #1:

The meta-research article of Cobey and colleagues presents a cross-sectional survey of biomedical researchers' perspectives on the reproducibility of research. The study builds on the widely cited Baker-survey published in Nature in 2016, but provides a more biomedical-centered perspective. In total, 1630 researchers successfully participated in this study. Briefly, 72% of participants agreed that there is a reproducibility crisis in biomedicine, with the leading cause of poor reproducibility being the 'pressure to publish' (which is interesting!). Only 16% of the participants indicated that their institution had established procedures to enhance the reproducibility of biomedical research; and 83% expected it to be harder to get funding for a replication study than for a novel one. Finally, the authors emphasize the relevance of the findings for policy makers and academic leadership to create incentives and research cultures that support better reproducibility and value research quality.

Overall, this is a short, but well-written manuscript on a highly relevant topic that presents the results of a survey on (ir)reproducibility in a clear and easy-to-read way. As stated by the authors themselves, the survey directly builds on the famous Nature-survey published in summer 2016, but differs in terms of the following two points: (1) the focus is exclusively on biomedical research (the Nature survey included scientists from various different disciplines across life sciences), (2) researchers were sampled from publication lists, allowing to report a response rate (this was not the case for the Nature survey). The latter point appears to be important to reduce any bias in the participants, as not only those researchers respond to the survey that are already actively involved in reproducibility projects. Whereas I agree on the importance of the latter point, I am not convinced by the overall novelty and design of the presented work. More specifically, I would like to raise the following major points:

(1) First of all, I am not entirely convinced by the novelty of the findings or the value of the outcome that goes beyond the Baker study. The fascinating aspect of the Baker study - at least from my perspective - was that so many researchers agreed on the existence of a crisis at a time (ir)reproducibility only started to become a hot topic in science. Furthermore, it was striking that the phenomenon was reported throughout the life sciences, arguing for a general rather than a discipline-specific problem. And finally, the Baker study provided a number of interesting causes for poor reproducibility that were discussed a lot and experimentally explored over the following years (thereby already creating the incentives for change that are discussed in the present paper). The findings of the present survey more or less confirm what has been reported previously and do not go beyond the already known problems. The focus on biomedical research is interesting, but not "groundbreaking". In case of publication, I would therefore like to see more reasons for the originality of this study as well as for its expected impact on the research community.

(2) The study is purely descriptive and the authors do not present any expectations or hypotheses. Although I agree that understanding researcher perceptions is key to driving a culture change and creating training interventions, one could also argue that the Baker study already met the criteria of an explorative study, allowing for generating some more concrete hypotheses for follow-up studies. For example, one could have looked at specific hypotheses surrounding the biomedical research community. What would you expect to be specific for this research field? Is it different from other disciplines within the life sciences? If so, why?

(3) It is not entirely clear to me, whether the survey was designed in such a way that it allows for reducing bias and identifying non-valid answers. I am not an expert in questionnaires' techniques, but I would expect to see some questions included that ask for the same thing, but with different words. From my perspective, such a validation step would be necessary to be able to extract only valid answers (and exclude those that were given simply to give any answer). The description of the survey in the Methods section appears to be rather limited in this respect. So, I would strongly recommend to provide more information about this.

(4) The discussion is rather short and does not go beyond the already known arguments. I therefore think that the manuscript could greatly benefit from a more in-depth analysis of the presented observations. In particular, it would be interesting to elaborate a bit more on the point that the perceived main reason was the pressure to publish. What is the expected impact of these findings and what kind of follow-up steps would be necessary to improve the situation? Is this specific for the biomedical research community?

Besides these major concerns, I do have a number of minor comments to the manuscript:

* The definition of reproducibility given in the introduction is a rather vague one. What does "consistent with the original study" mean? Is it dependent on the p-value (i.e. is a finding significant or not?) or should the observed effect simply go in a similar direction? Furthermore, what is meant by "similar methods"? How much variation is allowed? Is reproducibility here conceptualized as "generalizability"? Please be more specific, as this is the basis for the whole manuscript.

* At the end of the first section of the introduction the authors use the term "replicability"? Is it defined in a different way? Or is it used as synonym for reproducibility?

* The whole introduction is rather short, focusing on the Baker study and previous efforts to reproduce studies in different research fields. Whereas these are important aspects of the reproducibility topic, the issue is certainly much more complex and touches about various different aspects that could be mentioned here. For example, what are the discussed causes of poor reproducibility and what has been done to improve the situation? In particular, the topic of publication bias is an important one when it comes to systematic replication efforts (when trying to replicate published studies, you have to deal with the underlying publication bias that might be a reason for poor reproducibility in itself). I would therefore strongly recommend to expand the introduction a bit and include some more aspects that currently play a role in the debate.

* The methods are only superficially reported. For example, the section about the study design should also give more information about the different phases of the study (planning, piloting, surveying). Likewise, the section about the survey should provide more information about the technique used. How many questions were included? How was the survey organized? Were the questions presented in always the same or a randomized order? Did the authors include any kind of validation steps (see above)? Why did the authors modify some questions of the Nature study and others not?

* The survey was piloted by three researchers. Although I appreciate these piloting steps, I doubt the representativeness and generalizability of this step, as the involvement of just three scientists appears to be rather limited. Why only three? And what was changed according to the feedback of these researchers?

* The assignment of codes as reported in the data analysis section is not clear to me. What exactly was done here? What are potential consequences of the codes not being blinded?

* At the beginning of the results section it is mentioned that the authors obtained and used listed emails from all authors on an identified paper. What does it mean exactly? Did they include more than just one author per study? If yes, this could cause kind of pseudoreplication, as the authors of one study are more dependent on each other than the authors of different studies. Publications with a huge number of authors could therefore bias the outcome towards a specific direction. Please comment on that.

* The results are presented on the basis of pure numbers, completed by a huge number of tables. It would be much more readable to present some simplified figures or illustrations, as it was also done in the Baker study.

* The reference list is not formatted consistently. It is furthermore condensed to a surprisingly low number of references that could be complemented by several additional papers (related to the point that both the introduction and the discussion could cover additional topics).

* The statements concerning the link to the Baker-study are confusing. Sometimes it is explicitly stated that the aim was not to compare the data and a few sections later it is said that this approach allowed for comparing the results to the original study. Please be more concise on the aims and the actual relation to the Nature survey.

Reviewer #2:

[identified himself as Timothy M. Errington]

This manuscript describes a survey investigating biomedical researchers' views about reproducibility, including perceptions of the causes of irreproducibility, experiences with reproducibility, and perceived support for reproducibility from their institutions. The study builds off the 2016 Nature survey on this topic. Below are suggestions to help clarify details of the study and to improve accessibility to a broader readership.

1. There appears to be missing information. I could not find S1 (search strategy), S2 (full survey), S3 (breakdown of responses), and S4 (training resources)? I don't see it in the downloaded submission file or on the OSF project linked (though I did find this file that is maybe S2: https://osf.io/a3spd)? Related, the discussion refers to Table 7 as 'institutions influence on reproducibility of research' yet Table 7 is 'Thematic analysis of reasons why participants replicated their own study'. Finally, the authors share information about responses to a statement of "I feel I am more concerned about reproducibility than the average researcher in my field and at my career stage", yet I don't see anything in Table 9 about this question and responses?

2. The paper would benefit from presenting the tables in more easily digestible ways. For example, Table 3 presented two different ways is a bit confusing. I'd recommend picking one table to present instead of both (e.g., present numbers with percentages in parathesis for any given cell like Table 4). Another example, is for Table 4 it would be helpful if it was easy to digest. Maybe a heatmap?

3. Have the authors looked at any correlations of responses? For example, responses for if there is a 'crisis' (slight or significant) is at 72%, yet, when asked to give their perception of what proportion are reproducible responses are at the ~50% mark. A correlation analysis to understand the relationship between these two questions might be a valuable addition. Relatedly, what do the authors make of many responses being centered around the 'sometimes contributes' category in Table 4? Did the authors look to see if any respondents gave similar responses across the board on those questions? Or was there sufficient variation?

4. A potential limitation is the drop-off rate in the survey. It would great if the authors presented this information (graphically works well, but something to indicate where drop-off occurred). For example, there were 1630 respondents to at least one question, yet it's clear there was drop off since "A total of 724 participants responded to the item about why they replicated their own research".

5. A limitation section would be great - for example, the 'insufficient oversight' question of causes is limited in insight (and potential bias) by the sample being so heavily faculty/PI based (i.e., those responsible for oversight).

6. I see in table 6, the work 'replicate' and 'replication' are used? Similarly, in the discussion the authors switch feely between reproducibility and replicability. But the authors indicated they took a broad stance in the definition of reproducibility, which I took as inclusive of words like replicate and replication? What impact might this have on the responses given? For example, is the question "Have you ever tried to replicate a published study conducted by another team of authors and failed?". Respondents might have answered this as failed to get reproducible results or failed to be able to 'repeat the study exactly' as defined in the survey? This is different than question 8 in the survey (on OSF) where the failed is defined as "re-ran an experiment but got different results from the original study). Same for asking participants about whether another researcher contacted them who was 'unable to reproduce their findings', which based on definitions shared in the survey and manuscript would give a different response than if the authors asked 'unable to replicate their findings'? The terminology is indeed complex (as the authors acknowledge in the introduction - and thus there might be influence in the responses despite the authors best attempt at preventing it).

7. The discussion mentions 'reproducibility networks'. I agree on the need to mention, this, but just mentioning this and providing a reference I think is insufficient - it would benefit readers who have no idea what this is to give a little more context.

8. This EC study on reproducibility might be valuable to include in the discussion: https://op.europa.eu/en/publication-detail/-/publication/36fa41a8-dbd5-11ec-a534-01aa75ed71a1/language-en

9. Was the pilot testing with researchers include any of the authors of this manuscript? Same with the researchers who coded text response, were they any of the authors? It would be good to declare this either way.

10. Can you share your scripts (e.g., exported files form SPSS)? It would be nice to share these for others to review and reuse.

11. Can you share your anonymized data (or did you not get consent for that?).

Reviewer #3:

[identifies herself as Katherine S. Button]

This was an interesting and informative survey of biomedical researchers. The methods are generally sound and the findings are interesting. My only suggestions are that the results could be presented in a more interesting and informative manner using better data visualisation (graphs, figures, colour), and that the findings could be broken down by key categories (such as country, and career stage) to gain greater insights into international variation . 

Specific points below (page numbers would have helped): 

Methods: 

1) Piloting by three researchers seems a bit limited - a broader range including people outside of the research group might have been more rigorous. 

2) Analysis in SPSS - does this mean that the analysis code isn't available? Perhaps comment on the availability of meta-data and syntax/analysis code in the manuscript alongside data availability in the analysis section. Make it available if it currently is not. 

3) The thematic analysis method seems a bit thin. Consider citing some of the key thematic methods papers that informed your approach (e.g., Braun and Clarke) there are slightly varying approaches.

Results: 

Minor points: 

1) Determinants of irreproducibility: 97 (give %). Very small number - perhaps discuss as a limitation and it would be good to know who these people were. 

2) Provide all countries of employment in Table 1 to show how 'international' this survey is. The top few make it look very western-centric. 

Major points: 

1) I think the presentation of data in tables is very dry. My major suggestion for improving your paper is better data visualisation. Graphs, figures and colour would help. Think about people using your findings in presentations. These tables will look very dense on PowerPoint slides. 

2) Similarly, I think there is more analysis that can be done to look at variability across nations and career stages. I would suggest nice graphs/charts in the supplementary of the key findings broken down by nationality and career stage. Here in the UK for example we've had some movement re. institutional involvement and initiatives around reproducibility e.g., institutional members of UKRN. It would be interesting to see whether that comes across in the responses around intuitional support, for example. Also the reproducibility 'movement' has been readily taken up by early career researchers with grassroots initiatives led by PhD students and post-docs gaining international reach (e.g., Science RIOT club, ReproducibiliTea). So I'd be interested to see how these responses differ by career stage. Providing more fine-grained analysis along with the analysis code will help this survey to have its greatest impact. As above, helpful for people planning to use your analysis in presentations, discussions with their institutions ect. 

Check typos in penultimate paragraph. 

Discussion:

1) Depending on what you find in the additional analysis there might be key things to pull out in the discussion.

---

## [Decision Letter · Decision Letter 2]

22 Aug 2024

Dear Kelly,

Thank you for your patience while we considered your revised manuscript "Biomedical researchers’ perspectives on the reproducibility of research: a cross-sectional international survey" for publication as a Meta-Research Article at PLOS Biology. This revised version of your manuscript has been evaluated by the PLOS Biology editors, the Academic Editor and two of the original reviewers.

Based on the reviews, we are likely to accept this manuscript for publication, provided you satisfactorily address the remaining points raised by the reviewers, and the following data and other policy-related requests.

IMPORTANT - Please attend to the following:

a) Please truncate your Title to simply ""Biomedical researchers’ perspectives on the reproducibility of research"

b) Please address the concerns raised by reviewer #2. Like him, it was my impression that the OSF deposition is incomplete, and may not be sufficient to reproduce the study.

c) I note that you declare that you received no specific funding. Can you just confirm that this is correct?

d) When you mention your ethics approval in the MS (“This study received ethics approval from the Ottawa Health Sciences Research Ethics Board”), please can you also include the protocol approval number. From the files in OSF, it looks like this is #20210856-01H...

e) Please address my Data Policy requests below; specifically, we need you to supply the numerical values underlying Figs 1 and 2, either as a supplementary data file or as a permanent DOI’d deposition. I note that you already have an associated OSF deposition, but the values underlying these Figs look like they may already be presented in the Supplementary Tables...? Please ensure that everything is made available.

f) Please cite the location of the data clearly in all relevant Figure legends, e.g. “The data underlying this Figure can be found in Table S1” or “The data underlying this Figure can be found in https://osf.io/3ksvz/"

g) Please make any custom code available, either as a supplementary file or as part of your data deposition.

We expect to receive your revised manuscript within two weeks. 

*Published Peer Review History*

*Press*

Sincerely,

Roli

Roland Roberts, PhD

Senior Editor

rroberts@plos.org

PLOS Biology

DATA POLICY:

Regardless of the method selected, please ensure that you provide the individual numerical values that underlie the summary data displayed in the following figure panels as they are essential for readers to assess your analysis and to reproduce it: Figs 1 and 2. NOTE: the numerical data provided should include all replicates AND the way in which the plotted mean and errors were derived (it should not present only the mean/average values).

CODE POLICY

DATA NOT SHOWN?

REVIEWERS' COMMENTS:

Reviewer #1:

The authors made a good job in revising the manuscript. In particular, the discussion has improved a lot, covering a few more (novel) points and aspects. Thank you.

Reviewer #2:

[identifies himself as Timothy M. Errington]

Thank you for revising the manuscript. It is much improved from the previous version. The authors addressed my previous comments, however two minor aspects still remain.

1. While I appreciate the authors position on not wanting to do any correlations of the responses, it would benefit if the rationale given in the response to my comment was included in the manuscript somewhere. Not the response regarding a potential perception of the authors 'fishing' (which personally I would not think was happening since it would clearly labeled as exploratory and post-hoc), but more about the concern regarding being mindful of responses not being mutually exclusive with each other. This aspect could be added to the limitations section, especially since this would suggest caution for future researchers who might want to do this to look at between group differences.

2. I appreciate that the authors added their code and data to OSF, however I still can not view them. Maybe they are in a component that is still private?

---

## [Editor Report · Decision Letter 3]

1 Oct 2024

Dear Kelly,

Thank you for the submission of your revised Meta-Research Article "Biomedical researchers’ perspectives on the reproducibility of research" for publication in PLOS Biology. On behalf of my colleagues and the Academic Editor, Cilene Lino de Oliveira, I'm pleased to say that we can in principle accept your manuscript for publication, provided you address any remaining formatting and reporting issues. These will be detailed in an email you should receive within 2-3 business days from our colleagues in the journal operations team; no action is required from you until then. Please note that we will not be able to formally accept your manuscript and schedule it for publication until you have completed any requested changes.

Sincerely, 

Roli

Senior Editor

PLOS Biology

rroberts@plos.org